# The epidemiologic and economic impact of a quadrivalent human papillomavirus vaccine in Thailand

Wichai Termrungruanglert[1]*, Nipon Khemapech[1], Apichai Vasuratna[1], Piyalamporn Havanond[1], Preyanuch Deebukkham[2], Amit Sharad Kulkarni[3], Andrew Pavelyev[3]

1 Department of Obstetrics and Gynecology, Faculty of Medicine, Chulalongkorn University, King Chulalongkorn Memorial Hospital, Bangkok, Thailand, 2 MSD (Thailand) Ltd., Bangkok, Thailand, 3 Center for Observational and Real World Evidence, Merck & Co., Inc., Kenilworth, New Jersey, United States of America

* wichaiterm@yahoo.com

## Abstract

### Background

The human papillomavirus (HPV) vaccine was introduced into Thailand's national immunization program in 2017 for 11–12 year old school girls. The objectives of this study were to examine the epidemiological consequences and cost-effectiveness of a routine quadrivalent HPV (4vHPV) vaccination and the routine 4vHPV vaccination plus 5-year catch-up vaccination by comparing with cervical cancer screening only (no vaccination) in Thailand.

### Method

A transmission dynamic model was used to assess the cost-effectiveness of the routine 4vHPV vaccination and the routine 4vHPV vaccination plus catch-up vaccination, compared with no vaccination (screening only) in Thai population. The vaccination coverage rate assumptions were 95% in 11-12-year-old girls for the routine vaccination and 70% in 13–24 year-old females for the 5-year catch-up vaccination. Vaccination costs, direct medical costs of HPV-related diseases, and the number of quality of life years (QALYs) gained were calculated for over a 100-year time horizon with discount rate of 3%.

### Result

The model indicated that the routine 4vHPV vaccination and the routine plus catch-up 4vHPV vaccination strategies could prevent approximately 434,130 and 472,502 cumulative cases of cervical cancer, 182,234 and 199,068 cumulative deaths from cervical cancer and 12,708,349 and 13,641,398 cumulative cases of HPV 6/11 related genital warts, respectively, when compared with no vaccination over 100 years. The estimated cost per QALY gained (ICER) when compared to no vaccination in Thailand was 8,370 THB/QALY for the routine vaccination and 9,650 THB/QALY for the routine with catch-up vaccination strategy.

**Data Availability Statement:** All relevant data are within the paper and its Supporting Information files.

**Funding:** Article processing charge was provided by MSD (Thailand) Ltd.

**Competing interests:** The authors have read the journal's policy and the authors of this manuscript have the following competing interests: WT (corresponding author), NK, AV, and PH declare that they have no competing interests. PD was employee of MSD (Thailand) Ltd., during the study initiation and manuscript preparation. ASK and AP are employees of Merck & Co., Inc. These does not alter our adherence to PLOS ONE policies on sharing data and materials.

## Conclusion

Considering the recommended threshold of 160,000 THB/QALY for Thailand, the implementation of the routine 4vHPV vaccination either alone or plus the catch-up vaccination was cost-effective as compared to the cervical cancer screening only.

## Introduction

Human papillomavirus (HPV) infection is the most common sexually transmitted infection for both women and men, and can be passed through genital or skin-to-skin contact [1, 2]. HPV infection occurs almost immediately after becoming sexually active. Around half of these cases involve high-risk HPVs that are responsible for 5% of all cancers worldwide with 570,000 newly diagnosed female patients, and 60,000 newly diagnosed male patients annually [3] HPV vaccines, or HPV-virus-like particles (VLPs), effectively prevent HPV infections [4]. HPV not only causes cervical cancer but also many other cancer types including oropharyngeal cancers, anal cancer, penile cancer, vaginal cancer, and vulvar cancer. HPV is known to be responsible for between >60% (penile cancer) and almost 100% (cervical cancer) of these HPV-related cancers [5]. In 2018, Thailand reported 8,622 new cases of cervical cancer and 5,015 deaths [6], making it a leading cause of death in women as well as a national public health problem [6–8].

To address the public health problems and reduce mortality rates, Thailand introduced the HPV vaccine into the national immunization program in 2017 for grade 5 schoolgirls aged 11–12 years. The purpose of this study was to provide an overview of the results from a model developed to assess and test the public health and economic impact of quadrivalent HPV vaccination programs in Thailand. Specifically, this study aims to estimate:

1. The potential health effects and economic costs associated with the introduction of routine prophylactic vaccination with a 4vHPV vaccine [9, 10] in Thailand among 11–12 year old girls, along with cervical cancer screening. These health effects include those associated with the HPV types 6,11,16,18 related incidences of cervical intraepithelial neoplasia (CIN), cervical cancer, genital warts, and cervical cancer mortality in the population of Thailand.

2. The potential health effects and economic costs associated with the introduction of the routine vaccination among 11–12 year old girls along-with a catch-up vaccination of 13–24 year old females, and cervical cancer screening.

3. The cost-effectiveness of routine 4vHPV vaccination along with screening, and routine 4vHPV vaccination along with catch-up 4vHPV vaccination and screening, compared with no vaccination (cervical screening only).

## Materials and methods

A previously published transmission dynamic model by Elbasha & Dasbach [11] for the United States has been adapted for Thailand. The model estimates epidemiological and economic consequences of the use of the quadrivalent prophylactic vaccine against HPV 6/11/16/18 with cervical cancer screening. The model took into account both direct benefit to the vaccinated individuals and the benefit from herd immunity effect in the estimation. Individuals enter the model as they are born; move between successive age groups at an age and gender specific rate per year, and exit the model as they die.

## Target population

The target population for routine vaccination with 4vHPV is school girls aged 11–12 years old. The target population for catch-up vaccination is girls and women aged 13–24 years old. The impact will be estimated for the entire population of Thailand

## Scenarios

In the base case, the three scenarios being compared are: 1. Cervical cancer screening alone without vaccination (no vaccination), 2. Routine 4vHPV vaccination of girls aged 11–12 years old, along with the screening (routine 4vHPV vaccination), and 3. Routine 4vHPV vaccination of girls aged 11–12 years old plus catch-up vaccination of females 13–24 years old along with the screening (called "routine plus catch-up 4vHPV vaccination" in this article).

## Demographic and sexual mixing model inputs

The model developed simulates aging, all-cause mortality over time and HPV transmission within the Thai population through sexual mixing. Data on population size, age and gender specific all-cause mortality rates were taken from the Health Information Group, while data on sexual mixing were adapted from Elbasha & Dasbach [12]. Details on the sexual behaviors of the population and the parameters used in the model calibration are provided in S1 Table.

## Epidemiologic and clinical inputs

The age specific annual cervical cytology screening rates for Thailand were estimated based on Thai data from the 2009 Reproductive Health Survey [9], Center of Policy and Strategy of Ministry of Public Health (3rd National Survey on Thai Health 2004–2005) [13] and expert opinion (**Table 1**). In addition, based on the data from WHO/ICO Information Centre on HPV and Cervical Cancer 2010 [14], it was estimated that only 37.7% of Thai female population received adequate cervical cancer screening every year (defined as screening received once in 3 years).

We utilized data based on expert opinion (**Table 1**) for the percentage of all diagnosed cases of CIN 1 including mild dysplasia and condyloma acuminata (anogenital warts), CIN 2 which includes moderate dysplasia, and CIN 3 including severe dysplasia and carcinoma in situ (CIS) that are treated in Thailand. Data on other clinical parameters, such as sensitivity and specificity of cervical cytology and colposcopy tests, CIN treatment cure rates, persistence of HPV infections after treatment were estimated based on data from the literature review done by Insinga et al. [15]. Annual Thai age specific hysterectomy rates were estimated based on unpublished data from Department of Obstetrics and Gynecology, Faculty of Medicine, Chulalongkorn University and expert opinion (**Table 1**). Consistent with the range of values examined in previous cost-effectiveness studies of HPV vaccines [16–18], in our reference case analysis, we assumed lifetime duration of vaccine protection against HPV types 6/11/16/18.

## Quality of life measures

Quality-adjusted life years (QALYs) were estimated based on the following health utility measures: (a) Utilities for HPV disease states as reported for the U.S. model [11], and (b) Age and sex specific utility values for without HPV infection [20] (S1 Table).

## HPV vaccination strategies and characteristics

We assumed that the 4vHPV vaccination strategy will be combined with current cervical cancer screening and HPV disease treatment practices in Thailand. We examined the health and

**Table 1. Screening and treatment parameters.**

| Routine cervical cytology screening, excluding those with hysterectomy (% per year) | | |
|---|---|---|
| **Age group** | **Screening rate** | **Source** |
| 10–14 years | 0.00 | 1) Center of Policy and Strategy, Ministry of Public Health; 3rd National Survey on Thai Health 2004–2005 [13] |
| 15–19 years | 3.50 | 2) The 2009 Reproductive Health Survey, National Statistical Office Ministry of Information and Communication Technology of Thailand [9] |
| 20–24 years | 15.80 | 3) Expert opinion |
| 25–29 years | 30.00 | |
| 30–34 years | 28.30 | |
| 35–39 years | 38.24 | |
| 40–44 years | 40.50 | |
| 45–49 years | 42.69 | |
| 50–54 years | 40.24 | |
| 55–59 years | 33.86 | |
| 60–64 years | 15.00 | |
| 65–69 years | 10.00 | |
| 70–74 years | 5.00 | |
| 75–79 years | 3.0 | |
| 80–84 years | 2.00 | |
| 85 + years | 0.00 | |
| **Percent of female population that receives cervical cancer screening test at least once every 3 years** | | |
| Women adequately screened (% per year) | 37.7 | WHO/ICO 2010 [14] |
| Percent of women with a follow-up screening test following an abnormal PAP result | 68 | Sirisamutr et al 2012 [19] |
| **Recognize symptoms and seek treatment, % per year** | | |
| Localized cervical cancer | 2.6 (adjustable) | Elbasha & Dasbach 2010 [12] |
| Regional cervical cancer | 10 (adjustable) | Elbasha & Dasbach 2010 [12] |
| Distant cervical cancer | 90 (adjustable) | Elbasha & Dasbach 2010 [12] |
| **Proportion of CIN/CIS is treated, by stage (%)** | | |
| Percent of CIN 1 treated | 40 | Expert opinion |
| Percent of CIN 2 treated | 75 | |
| Percent of CIN 3 treated | 75 | |
| Percent of CIS treated | 75 | |
| **Proportion of genital warts cases treated (%)** | | |
| Percent of genital warts treated | 40 | Expert opinion |
| **Hysterectomy for non–HPV-related conditions rate (per year)** | | |
| **Age Groups** | **Rate** | **Source** |
| 15–24 years | 0.02 | Unpublished Data: Department of Obstetrics and Gynecology, Faculty of Medicine, Chulalongkorn University plus expert opinion |
| 25–29 years | 0.26 | |
| 30–34 years | 0.53 | |
| 35–39 years | 0.89 | |
| 40–44 years | 1.17 | |
| 45–54 years | 0.99 | |
| 55+ years | 0.36 | |

\* CIN, cervical intraepithelial neoplasia; CIS, carcinoma in situ

economic impacts of three different scenarios: 1) no vaccination, 2) routine 4vHPV vaccination; and 3) routine plus catch-up 4vHPV vaccination. All scenarios were evaluated assuming that the current cervical cancer screening would remain at a constant rate throughout the time horizon of the evaluations. Based on the local experience with other vaccines in Thailand [21], we assumed that for the routine 4vHPV vaccination, 95% of all girls 11–12 years of age received the routine regimen. For the catch-up vaccination, it was assumed that 70% of females between 13–24 years were vaccinated in the first 5 years. Girls younger than 13 received 2 doses, while older females received three doses.

## Economic data

We adopted a healthcare perspective to assess costs. The costs were broken down into 3 categories including costs of vaccination, cervical cancer screening and costs of treatment in Thailand. Cost data and corresponding references can be found in S1 Table. We assumed a cost of 500.00 THB per vaccine dose ($\approx$ US$15) for each vaccine that was administered. The price in USD was estimated based on conversion rate of 1 USD /34.35 THB from http://finance.yahoo.com/currency-converter/#from=USD;to=THB;amt=1 accessed on April 3rd 2017. The cost per vaccine dose used was the cost proposed to pay in 2012 by the Ministry of Public Health of Thailand [22, 23]. Non-medical costs (e.g. transportation) and indirect costs (e.g. productivity loss) were not included in the analysis. Costs and health benefits (QALYs) were discounted by 3% [24].

## Model outputs

We used a number of outcome measures to assess the epidemiologic impact and cost-effectiveness of each vaccination strategy. Epidemiologic outputs included CIN 2/3, CIN 1, invasive cervical cancer, genital warts cases for females and males, and cervical cancer related deaths. The economic outputs of interest from the model included total costs, quality-adjusted life years, and incremental cost per QALY. We measured the cost per QALY ratio as the incremental cost difference between the two strategies divided by the incremental QALY difference between the two strategies (i.e. Incremental Cost Effectiveness Ratio, ICER).

## Model calibration

We assessed the predictive validity of the model by calibrating model predictions to the observed data of the incidence and mortality of cervical cancer in Thailand that was attributable to HPV 16/18. We also calibrated the model to the incidence rate of genital warts estimated from available data.

The total crude cervical cancer incidence rate in Thailand was reported as 29.2 per 100,000 [25] of which approximately 73.8% [14], i.e. $\approx$ 21.55 per 100,000, was assumed to be attributable to HPV 16/18. The corresponding crude cervical cancer mortality rate was reported as 12.7 per 100,000 [25] of which approximately 73.8% [14], i.e. 9.37 per 100,000, was assumed to be attributable to HPV 16/18. The model was calibrated using cervical cancer recognition and transmission probability parameters. At the beginning of the calibration process, all parameters were set to the values taken from Elbasha & Dasbach [12]. Then the cervical cancer recognition parameters were adjusted to obtain the observed cervical cancer incidence rate to cervical cancer mortality rate ratio (i.e. 21.55/9.37 $\approx$ 2.30). After that we adjusted the per sexual partnership transmission probability of HPV 16/18. Incidence rates for genital warts were 231 per 100,000 [8]. Approximately 90%, i.e. $\approx$ 208 per 100,000, were attributable to HPV 6/11.

## Sensitivity analyses

Sensitivity analyses were conducted to examine how changes in the vaccination programs (routine vaccination and routine plus catch-up vaccination) or health-related benefit (health-related benefits, specifically HPV6/11-related genital lesions with and without HPV 6/11 related benefits) influenced the estimated cost-effectiveness ratio (Cost/QALY).

## Horizon

Analysis was performed for 100-year time horizon, with 5 years interval to allow for the complete health and economic vaccination impact, in alignment with Elbasha et al. [11].

## Cost effectiveness analysis

To assess the cost effectiveness of the vaccination strategy in preventing the disease, the model estimated the total discounted costs and effects (i.e., QALYs) accrued over the 100 year time horizon for both vaccine strategies that were being evaluated (i.e. no vaccination scenario, the routine vaccination of 11–12 year old girls, and routine vaccination of 11–12 year old girls plus the catch-up vaccination of 13–24 year old females). Next, the model calculated the incremental cost incurred to achieve the incremental benefit from vaccination. This was then used to calculate the ICER (the ratio of the incremental costs to incremental QALYs gained).

# Results

## Public health impact of the HPV vaccination strategies

**Impact on the HPV6/11/16/18-related diseases.**   The routine 4vHPV vaccination and the routine plus catch-up 4vHPV vaccination strategies would cumulatively prevent cervical cancer for 434,130 cases (62.1%) and 472,502 cases (67.6%) over 100 years respectively when compared to no vaccination. The model predicted that HPV16/18 related deaths over 100 years would be reduced by 182,234 cases (59.6%) and 199,068 cases (65.1%) for the routine 4vHPV vaccination and the routine plus catch-up 4vHPV vaccination strategies, respectively when compared to no vaccination.

The routine 4vHPV vaccination would also cumulatively prevent 1,984,634 cases of HPV16/18 related CIN1, 4,045,772 cases of HPV16/18 related CIN2/3, 533,677 cases of HPV6/11 related CIN1 and 12,708,349 cases of HPV6/11 related genital warts among women and men over 100 years, compared with no vaccination. Case reductions for the routine plus catch-up 4vHPV vaccination were higher than that for routine 4vHPV vaccination. The routine plus catch-up vaccination would also cumulatively prevent 2,133,772 cases of HPV16/18 related CIN1, 4,346,356 cases of HPV16/18 related CIN2/3, 574,097 cases of HPV6/11 related CIN1 and 13,641,398 cases of HPV6/11 related genital warts among women and men compared with no vaccination. **Table 2** shows the cumulative case and percent reduction of these outcomes over 0, 5, 25, and 100 years. (**S1 Fig** provides the graphical presentations of the estimated number of the disease events in the routine vaccination and the routine plus catch-up vaccination in a population of 100,000 over 100 years)

**Impact on healthcare cost.**   At population level, the routine vaccination and the routine plus catch-up vaccination strategies were projected to avoid the direct medical costs of the HPV-related diseases for 57,471,932,088 THB and 69,487,905,607 THB over 100 years when compared to no vaccination, respectively. Approximately three-fourths of the treatment costs avoided were from HPV16/18-related treatment costs (77.2% for routine vaccination and 77.0% for routine plus catch-up vaccination) and one-fourth were HPV6/11-related treatment costs (22.8% and 23.0%, correspondingly). (**S2A and S2B Fig** present the estimated healthcare

**Table 2. Estimated cumulative reductions of the HPV-related disease outcomes at Thai population level over 100 years.**

| Health related outcome | Routine 4vHPV vaccination vs no vaccination | | | | Routine and catch-up 4vHPV vaccination vs no vaccination | | | |
|---|---|---|---|---|---|---|---|---|
| | 5 yrs | 25 yrs | 50 yrs | 100 yrs | 5 yrs | 25 yrs | 50 yrs | 100 yrs |
| **Cumulative case reduction[a]** | | | | | | | | |
| HPV16/18 related deaths | 0 | 1932 | 38212 | 182234 | 1 | 5273 | 5237 | 199068 |
| Cervical cancer | 0 | 8060 | 102342 | 434130 | 16 | 18638 | 135675 | 472502 |
| HPV16/18 CIN1 | 106 | 143844 | 706943 | 1984634 | 4262 | 254545 | 853427 | 2133722 |
| CIN2/3 | 198 | 299419 | 1449621 | 4045772 | 81183 | 521514 | 1734445 | 4346356 |
| HPV6/11 Genital warts | | | | | | | | |
| In female | 4837 | 867965 | 2679925 | 6372998 | 106799 | 1324470 | 3156476 | 6849994 |
| In male | 1490 | 725258 | 2536286 | 6335351 | 51514 | 1137032 | 2990177 | 6791404 |
| HPV6/11 related CIN1 | 78 | 60061 | 214666 | 533677 | 3980 | 97495 | 255033 | 574097 |
| **Cumulative % reduction[b]** | | | | | | | | |
| HPV16/18 related deaths | 0 | 2.5 | 25.0 | 59.6 | 0.0 | 6.9 | 34.3 | 65.1 |
| Cervical cancer | 0 | 4.6 | 29.3 | 62.1 | 0.0 | 10.7 | 38.8 | 67.6 |
| HPV16/18 CIN1 | 0.1 | 22.0 | 54.2 | 76.1 | 3.3 | 39.0 | 65.4 | 81.8 |
| CIN2/3 | 0.1 | 22.5 | 54.1 | 76.0 | 3.1 | 39.2 | 65.1 | 81.6 |
| HPV6/11 Genital warts | | | | | | | | |
| In female | 1.3 | 47.0 | 72.5 | 86.3 | 28.9 | 71.7 | 85.5 | 92.7 |
| In male | 0.4 | 38.1 | 66.7 | 83.3 | 13.4 | 59.8 | 78.6 | 89.3 |
| HPV6/11 related CIN1 | 0.2 | 37.6 | 67.3 | 83.6 | 12.5 | 61.1 | 79.9 | 90.0 |

a. Cases rounded to nearest 1.

b. Percentages round to nearest 0.1

costs avoided over 100 years by HPV genotypes at population levels of the routine vaccination and the routine plus catch-up vaccination when compared to no vaccination.) The estimated vaccination costs over 100 years were 17,108,393,827 THB and 24,943,313,705 THB for the routine vaccination and the routine plus catch-up vaccination respectively. (**Table 3** and **S2 Fig**)

**Table 3. Estimated costs over 100 years at population level.**

| Cost | Estimated cumulative costs over 100 years (THB) | | |
|---|---|---|---|
| | No vaccination (screening only) | Routine 4vHPV vaccination | Routine and catch-up 4vHPV vaccination |
| **Direct medical costs of HPV-related diseases treatment** | | | |
| Cervical cancer | 77,212,923,754 | 52,312,413,664 | 47,143,360,147 |
| HPV16/18 CIN1 | 1,774,039,845 | 908,082,629 | 728,196,733 |
| CIN2/3 | 38,293,570,197 | 19,706,623,671 | 15,922,378,541 |
| HPV6/11 Genital warts | | | |
| In male | 10,283,135,581 | 4,020,431,254 | 2,692,431,841 |
| In female | 9,986,580,109 | 3,366,656,208 | 1,864,840,212 |
| HPV6/11 related CIN1 | 396,580,852 | 160,732,517 | 107,751,083 |
| **Total direct medical costs of HPV-related diseases treatment** | **137,946,830,338** | **80,474,939,944** | **68,458,958,556** |
| HPV-related disease costs avoided* | | 57,471,932,088 | 69,487,905,607 |
| **Vaccination cost** | 0 | 17,108,393,827 | 24,943,313,705 |

* Net when compared to no vaccination.

**Table 4. Estimated cost per QALY gained by adding routine 4vHPV vaccination or routine plus catch-up 4vHPV vaccination to cervical cancer screening in Thailand.**

| | Discounted Total | | Incremental | | |
|---|---|---|---|---|---|
| | Costs/Person (THB) | QALYs/Person | Costs/Person (THB) | QALYs/Person | Costs/QALYs (THB) |
| **Base case** | | | | | |
| Screening only (no vaccination) | 30,183.6 | 28.53312 | – | – | – |
| Routine 4vHPV vaccination | 30,520.9 | 28.59342 | 337.28 | 0.04029 | 8,370 |
| Routine plus catch-up 4vHPV vaccination | 30,660.1 | 28.60250 | 476.52 | 0.04938 | 9,650 |
| **Sensitivity analysis** | | | | | |
| *Base case with excluding HPV 6/11 health related disease* | | | | | |
| Screening only (no vaccination) | 29,836.8 | 28.55760 | – | – | – |
| Routine 4vHPV vaccination | 30,404.1 | 28.59511 | 540.28 | 0.03751 | 14,420 |
| Routine plus catch-up 4vHPV vaccination | 30,587.9 | 28.60458 | 724.13 | 0.04599 | 15,747 |

Costs rounded to 0.01, QALYs rounded to 0.00001, and Costs/QALYs rounded to 1

Comparing to screening only strategy.

$$\text{ICER} = \frac{\textit{Costs in the vaccination Strategy} - \textit{Cost in screening only}}{\textit{QALY in the vaccination Strategy} - \textit{QALY in screening only}}$$

### Cost-effectiveness of HPV vaccination strategy

**Base case analysis.** Under base-case analysis, the estimated incremental costs per person by adding the routine 4vHPV vaccination and the routine plus catch-up 4vHPV vaccination, compared with no vaccination, were 337.28 THB and 476.52 THB, respectively and the estimated QALY gained were 0.040 and 0.049, respectively. The estimated cost per QALY gained by adding the routine 4vHPV vaccination and the routine plus catch-up vaccination to the existing cervical cancer screening were 8,370 and 9,650 THB, respectively (**Table 4**).

**Sensitivity analysis.** We conducted a variety of sensitivity analyses to test the effect of certain parameters on the incremental cost-effectiveness ratios (ICER). Excluding of HPV 6/11 related benefits resulted in an increase of the estimated cost per QALY gained by 72.3% for routine vaccination programs and by 63.2% for routine plus catch-up program.

**Model calibration and validation.** The model predicted with the current screening, in the absence of vaccination, an annual HPV16/18 related cervical cancer incidence of about 21.6 per 100,000 females compared to 21.55, (i.e. with a relative error of approximately 1%). In addition, the model predicted an annual HPV 16/18 related cervical cancer mortality of about 9.45 per 100,000 females compared to 9.37, (i.e. with a relative error of less than 1%). The model also predicted the overall incidence of HPV 6/11 related genital warts to be about 229 per 100,000 per year in females and about 235 per 100,000 per year in males compared to 208 and 208 per 100,000 correspondingly.

## Discussion

We examined the health and economic impact of 4vHPV vaccination in Thailand by adapting a previously developed transmission dynamic model [11]. The cost of vaccine used in this study was the cost proposed to pay by the Ministry of Public Health of Thailand in 2012 in order to incorporate the HPV vaccine into the national program [22, 23]. This vaccine cost was lower than the selling price by the private hospitals in Thailand but higher than the willingness to pay by the parents if it was not offered for free [26]. The prices of HPV vaccines were declined with the passage of time due to many reasons, e.g. competitive tendering of prices [27, 28], the involvement of organizations, such as the Global Alliance for Vaccine

Initiative (GAVI), the World Health Organization, United Nations International Children's Emergency Fund (UNICEF), and the World Bank and the Bill & Melinda Gates Foundation that were established to help children worldwide, to improve the access to vaccines [27, 28], and having more suppliers more competition [27]. The affordable prices of vaccines could be achieved through tender procedures because vaccine pricing is affected by contract volume and its duration, country, per capita gross domestic product (GDP), and the number of offers received [27, 28].

The results indicated that introducing a national immunization program with a 4vHPV vaccine (types 6/11/16/18) for girls aged 11–12 years in Thailand may reduce the incidence and cost associated with HPV 6/11/16/18 related cervical cancer, precancerous lesion and genital warts in Thailand. All cases avoided in the first few years of the vaccination program were due to reduction in the incidence of genital warts and CIN. The effect of the routine vaccination strategy as well as routine plus catch-up vaccination was to steadily reduce the incidence of HPV16/18 related cervical cancer cases and deaths. Compared to the routine vaccination, the routine plus catch-up program demonstrates a higher reduction in the incidence rates of HPV related disease.

The reduction of HPV6/11 related genital warts cases in male results from the herd immunity benefits of female vaccination and shares similar qualitative features with those of cervical cancer. However, because CIN 2/3, CIN 1, and genital warts occur sooner following HPV infection, the curves are shifted to the left compared with the cervical cancer curves. These shifts mean that the reduction in genital warts, CIN1, and CIN2/3 will occur sooner than cervical cancer after the implementation of HPV vaccination which has a positive impact on health care benefits. This shift was most pronounced in the curves for HPV 6/11 related CIN1 and genital warts (S1C and S1D Fig). In both cases about 77% of the avoided costs was due to the reduction in HPV16/18 related treatments over the 100- year period (S2A and S2B Fig). Vaccination results in reductions in direct medical cost of HPV related disease, which are greater than the increase in vaccine costs. However, the incremental cost is still higher for vaccination scenarios because of the increases in cost of screening, which increases with increases in life expectancy.

WHO recommends the use of the gross domestic product (GDP) per capita as thresholds to derive the following three categories of cost effectiveness: highly cost effective (less than one GDP per capita), cost effective (between one and three times GDP per capita), and not cost-effective (more than three times GDP per capita) [29]. In 2014 the Thailand threshold for cost effectiveness was about 160,000 THB/QALY [30]. Therefore, the incremental cost per QALY (ICER) estimated by the model to be 8,370 THB/QALY indicates that routine 4vHPV vaccination is a highly cost effective strategy in Thailand. In the sensitivity analyses, the cost-effectiveness of the 11–12 year old female vaccination strategy in Thailand was relatively robust.

## Limitations

The results of this analysis should be interpreted with the following limitations. Firstly, there is a limitation in the availability of Thai specific data in a key number of parameters, including data on the incidence of genital warts, CIN1, and CIN2/3 in the general population, therefore we could not calibrate the model to these outcomes. The model was calibrated specifically to cervical cancer incidences rates, mortality rates and genital warts and hence is conservative as no other cancers (such as vulvar, vaginal, anal, head and neck and penile cancers) were taken into consideration. Secondly, only direct medical costs were included. This potentially underestimates the benefits of vaccination. In this case, we used the default model parameters. In addition, the computation of ICER (Cost/QALY gained) is based on direct cost medical variables. Indirect medical cost and other non-medical economic variables are not used.

## Conclusions

In conclusion, our study demonstrates that the routine quadrivalent HPV vaccination of 11–12 years girls can be a highly cost effective intervention in Thailand that can substantially reduce the burden of cervical diseases and genital warts. Introduction of the 5 year catch-up program for 13–24 year old females along with the routine vaccination of 11–12 year old girls has an even higher impact on the reduction of cervical diseases and genital warts.

## Supporting information

**S1 Fig. Estimated number of HPV-related disease events over 100 years in the routine 4vHPV vaccination, routine plus catch-up 4vHPV vaccination, compared to no vaccination (cervical cancer screening only) in a Thai population of 100,000 (base case analysis). A,** Estimated HPV 16/18 Related Incidence of Cervical Cancer Among Females over 100 Years; **B**, Estimated HPV 16/18 Related Cervical Cancer Deaths Among Females over 100 years; **C**, Estimated HPV 16/18 Related incidence of CIN1 Among Females over 100 years; **D**, Estimated HPV 16/18 Related Incidence of CIN 2/3 Among Females over 100 years; **E,** Estimated HPV 6/11 Related Incidence of CIN1 Among Females over 100 years; **F**, Estimated HPV 6/11 Related Incidence of Genital Warts Among Females over 100 years.
(DOCX)

**S2 Fig. Estimated HPV-related treatment costs avoided over 100 years by HPV genotypes in Thailand when compared to no vaccination (screening only). A**, For routine 4vHPV vaccination; and **B**, For routine plus catch-up 4vHPV vaccination.
(DOCX)

**S1 Table. All-cause mortality and cervical cancer mortality.**
(DOCX)

**S2 Table. Sexual behavior parameters.**
(DOCX)

**S3 Table. Quality-of-life parameters.**
(DOCX)

**S4 Table. Costs of diagnosing and treating HPV disease in Thailand (THB).**
(DOCX)

## Author Contributions

**Conceptualization:** Wichai Termrungruanglert.

**Data curation:** Wichai Termrungruanglert, Nipon Khemapech, Apichai Vasuratna, Piyalamporn Havanond, Preyanuch Deebukkham.

**Formal analysis:** Amit Sharad Kulkarni, Andrew Pavelyev.

**Investigation:** Amit Sharad Kulkarni, Andrew Pavelyev.

**Methodology:** Wichai Termrungruanglert.

**Software:** Amit Sharad Kulkarni, Andrew Pavelyev.

**Supervision:** Wichai Termrungruanglert.

**Validation:** Wichai Termrungruanglert, Piyalamporn Havanond, Preyanuch Deebukkham.

**Writing – original draft:** Wichai Termrungruanglert, Nipon Khemapech, Apichai Vasuratna, Piyalamporn Havanond.

**Writing – review & editing:** Wichai Termrungruanglert, Nipon Khemapech, Apichai Vasuratna, Piyalamporn Havanond, Preyanuch Deebukkham.

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
