## [Decision Letter · Decision Letter 0]

9 Nov 2020

PONE-D-20-29823

The Epidemiologic and Economic Impact of a Quadrivalent Human Papillomavirus Vaccine in Thailand.

PLOS ONE

Dear Dr. Termrungruanglert,

Thank you for submitting your manuscript to PLOS ONE. After careful consideration, we feel that it has merit but does not fully meet PLOS ONE’s publication criteria as it currently stands. Therefore, we invite you to submit a revised version of the manuscript that addresses the points raised during the review process.

We look forward to receiving your revised manuscript.

Kind regards,

Luca Giannella

Academic Editor

PLOS ONE

Journal Requirements:

"I have read the journal's policy and the authors of this manuscript have the following competing interests:

Article processing charge (if any) was provided by MSD (Thailand) Ltd.

WT (corresponding author), NK, AV, and PH declare that they have no competing interests.  PD was employee of MSD (Thailand) Ltd., during the study initiation and manuscript preparation.  ASK and AP are employees of Merck & Co., Inc"

5. Please upload a copy of Figure 1 to which you refer in your text on page 14. If the figure is no longer to be included as part of the submission please remove all reference to it within the text.

Reviewers' comments:

Reviewer's Responses to Questions

**Comments to the Author**

1. Is the manuscript technically sound, and do the data support the conclusions?

Reviewer #1: Partly

Reviewer #2: Yes

2. Has the statistical analysis been performed appropriately and rigorously? 

Reviewer #1: I Don't Know

Reviewer #2: Yes

3. Have the authors made all data underlying the findings in their manuscript fully available?

Reviewer #1: No

Reviewer #2: Yes

4. Is the manuscript presented in an intelligible fashion and written in standard English?

Reviewer #1: Yes

Reviewer #2: Yes

5. Review Comments to the Author

Reviewer #1: It’s an interesting work relating to the epidemiological consequences and cost-effectiveness of the HPV vaccine. In this study, the cost-effect analysis was used to evaluate the cost and the health benefits of HPV vaccination among all eligible women in Thailand. However, there are some questions that need to be confirmed and revised, which is associated with the conclusions of the study.

The key points of this study might be the cost of the HPV vaccine and the cost of cervical cancer screening. However, the price of the vaccine is cheaper than in other countries, and the baseline of the screening is longer than the guidelines.

Major

1. “We assumed a cost of 500.00 THB 1 per vaccine dose (≈US$15) for each vaccine that was administered”

Is the price of 4vHPV vaccination so cheap? Generally speaking, the price of 4vHPV vaccination per vaccine dose is US$125. I think this price needs to be confirmed.

2. The recommended screening for cervical cancer is HPV plus TCT liquid-based cytology once a year. The price is around US$75.

What medical screening method is used in your model, and what is the price and frequency of your screening?

3. Cervical cancer treatment covers surgery, chemotherapy, and radiotherapy, which cost about US$20000

I think your data is a bit biased for the above three key costs. I hope that after you have confirmed these costs, you can perform the model calculation again based on the actual situation in your country.

Minor

1. “In 2012 alone, Thailand reported 8,184 new cases of cervical cancer and 4,513 deaths”.

It is best to update to the latest data

2. “Costs and health benefits (QALYs) were discounted by 3%”.

What’s the meaning of “discounted by 3%”?

Reviewer #2: The study used a transmission dynamic model to assess the cost-effectiveness of the routine 4vHPV vaccination and the routine 4vHPV vaccination plus catch-up vaccination, compared with no vaccination (screening only) in Thai population,which is very meaningful. There are two issues that need to be explained.

1.In HPV Vaccination Strategies and Characteristics, the author described “Based on the local experience with other vaccines in Thailand, we assumed that for the routine 4vHPV vaccination, 95% of all girls 11-12 years of age received the routine regimen. For the catch-up vaccination, it was assumed that 70% of females between 13-24 years were vaccinated in the first 5 years”, is there any corresponding data to support the assumed percentage of vaccination?

2.In Model Calibration, the author described “the cervical cancer recognition parameters were adjusted to obtain the observed cervical cancer incidence rate to cervical cancer mortality rate ratio (i.e. 21.55/11.22 ≈ 2.30) “, but “the corresponding crude cervical cancer mortality rate was reported as 12.7 per 100,000 of which approximately 73.8%, i.e. 9.37 per 100,000, was assumed to be attributable to HPV 16/18. “, is there a calculation error in this value?

6. PLOS authors have the option to publish the peer review history of their article (what does this mean?). If published, this will include your full peer review and any attached files.

Reviewer #1: No

Reviewer #2: No

---

## [Author Response · Author response to Decision Letter 0]

16 Dec 2020

14 DEC 2020

Dear Editor,

RE: Revision of Manuscript ID# PONE-D-20-29823 (1st revision)

We would like to thank you and the reviewers for your review and comments dated 09- November 9, 2563 and for the opportunity to revise our manuscript entitled “The Epidemiologic and Economic Impact of a Quadrivalent Human Papillomavirus Vaccine in Thailand.” (Manuscript ID PONE-D-20-29823). We appreciate the time and efforts by the editor and reviewers in reviewing this manuscript.

Based on the instructions/comments provided, we uploaded the file of the revised manuscript on the journal website.

We have revised the manuscript by reformatting the manuscript and modifying the Materials and Methods, Discussion sections, and Supplementary Table 4 based on the comments. Accordingly, we have uploaded a copy of the original manuscript marked with all the changes made during the revision process (Tracked changes). The new text is underlined while the crossed-out text refers to the deleted original text.

Enclosed to this letter is our point-by-point response to the comments. As you notice, we agreed with all the comments raised by the reviewers. We would like to take this opportunity to express our sincere thanks to the editor and reviewers of our manuscript. We would like also to thank you for allowing us to resubmit a revised copy of the manuscript.

I hope that the revised manuscript is accepted for publication in PLOS ONE.

Sincerely,

Wichai Termrungruanglert, M.D. (Corresponding author) 

Department of Obstetrics and Gynecology, 

Faculty of Medicine, Chulalongkorn University, 

King Chulalongkorn Memorial Hospital, 

Bangkok, Thailand. 

Phone: 6681-9333546; E-mail: wichaiterm@yahoo.com

 

Editor Comments to Author:

Authors’ Response: We reformatted the manuscript according to PLOSE ONE style. 

"I have read the journal's policy and the authors of this manuscript have the following competing interests: Article processing charge (if any) was provided by MSD (Thailand) Ltd. WT (corresponding author), NK, AV, and PH declare that they have no competing interests. PD was employee of MSD (Thailand) Ltd., during the study initiation and manuscript preparation. ASK and AP are employees of Merck & Co., Inc"

Authors’ Response: We confirmed that "These do not alter our adherence to PLOS ONE policies on sharing data and materials” and added the confirmation sentence after the previous competing interests statement and the cover letter. 

Authors’ Response: The captions were added at the end of manuscript. 

Authors’ Response: Amendment done. 

5. Please upload a copy of Figure 1 to which you refer in your text on page 14. If the figure is no longer to be included as part of the submission please remove all reference to it within the text.

Authors’ Response: The figure is “Supplementary Figure 1”. 

Review Comments to the Author

Reviewer 1

 It’s an interesting work relating to the epidemiological consequences and cost-effectiveness of the HPV vaccine. In this study, the cost-effect analysis was used to evaluate the cost and the health benefits of HPV vaccination among all eligible women in Thailand. However, there are some questions that need to be confirmed and revised, which is associated with the conclusions of the study.

 The key points of this study might be the cost of the HPV vaccine and the cost of cervical cancer screening. However, the price of the vaccine is cheaper than in other countries, and the baseline of the screening is longer than the guidelines.

Major

1. “We assumed a cost of 500.00 THB 1 per vaccine dose (≈US$15) for each vaccine that was administered”. Is the price of 4vHPV vaccination so cheap? Generally speaking, the price of 4vHPV vaccination per vaccine dose is US$125. I think this price needs to be confirmed.

Authors’ Response: 

 Cost of the vaccine used in this study was the cost that the Ministry of Public Health Thailand proposed to pay in 2012 in order to incorporate the HPV vaccine into the national program [1, 2]. Although the cost used in the model was lower than the HPV vaccine prices at Public hospitals, however, the cost in the study still has been higher than the willingness to pay by the parents if it was not offered for free. A survey of willingness to copay showed that the parent would copay less than 500 THB for three doses of HPV vaccine and no significant difference between bivalent and quadrivalent vaccines for willingness to pay was identified [3]. We additionally described the cost of vaccine that we used in Method (Economic Data) and Discussion. The references are also provided in both manuscript and Supplementary Table 4. 

 At the time of the launch, the first HPV vaccines were by far the most expensive of all available vaccines in the market, having a cost of USD 120.11 (EUR 100) per dose in most European countries. However, the prices of vaccines were declined with the passage of time probably due to maturation in price[4]. For example, the dose price of HPV vaccine funded by Pan American Health Organization Revolving Fund (PAHO-RF) declined from USD 32.00 in 2010 to USD 13.50 in 2013 and then to USD 8.50 in 2015 [4]. The reasons for the drop in vaccine prices can be many reasons, e.g. competitive tendering of prices in the case of HPV vaccines[4] [5], the involvement of organizations such as the Global Alliance for Vaccine Initiative (GAVI), the World Health Organization, the UNICEF, the World Bank and the Bill & Melinda Gates Foundation, and the PAHO-RF. For example, in 2000, the GAVI was established to help children worldwide, to improve the access to vaccines and support the vaccinations in 40 countries for 30 million girls by 2020 and has improved the affordability of vaccines, particularly for low-income countries[4] [5]. GAVI-funded vaccines promoted more suppliers into the market and more competition, thus helping to lower the price of some vaccines[5].

 It was observed that the affordable prices of vaccines can be achieved through tender procedures[4] [5]. This is because vaccine pricing is affected by contract volume and its duration, country, per capita GDP and the number of offers received [5]. This could also be explained by economic theory which suggests that with higher purchasing power on the demand side, the price tendering may achieve significant savings. Therefore, the cost of 500 THB/dose is a possible cost.

2. The recommended screening for cervical cancer is HPV plus TCT liquid-based cytology once a year. The price is around US$75. 

What medical screening method is used in your model, and what is the price and frequency of your screening?

Authors’ Response: The cervical screening method in the model is annual screening. The method includes cytology plus HPV test. The cost (1876 THB � 55 USD) is the estimated cost of the cervical cytology screening method using the cost published by Sharma M, et al 2012 [6]. The reference has been cited in Supplementary Table 4.

3. Cervical cancer treatment covers surgery, chemotherapy, and radiotherapy, which cost about US$20000. I think your data is a bit biased for the above three key costs. I hope that after you have confirmed these costs, you can perform the model calculation again based on the actual situation in your country.

Authors’ Response: We confirmed that the cost used in the model is the actual cost in Thailand. (Data from King Chulalongkorn Memorial Hospital, Bangkok, Thailand).

Minor

1. “In 2012 alone, Thailand reported 8,184 new cases of cervical cancer and 4,513 deaths”. It is best to update to the latest data

Authors’ Response: Thank you for your suggestion. We updated the data using the information in 2018. “In 2018 alone, Thailand reported 8,622 new cases of cervical cancer and 5,015 deaths”[7].

2. “Costs and health benefits (QALYs) were discounted by 3%”. What’s the meaning of “discounted by 3%”?

Authors’ Response: The costs and benefits often considered in a health economic evaluation or health technology assessment (HTA) are not only incurred in the current year, but materialize beyond the present[8]. For the valuation of costs and benefits in the context of an economic evaluation, their timing is relevant because people generally value future costs and effects less than current costs and effects and their value diminishes the more distant in the future they occur. Hence, economic evaluations need to adjust the value of costs and benefits for the time at which they occur, a technique known as discounting[8]. It is a common practice in health economic evaluations to perform discounting on both future costs and benefits[9]. A review of relevant case studies and guidelines and provide guidance for all researchers conducting economic evaluations of health technologies in the Thai context. A uniform discount rate of 3% is recommended for both costs and health effects in base case analyses [9]. 

Reviewer #2: 

The study used a transmission dynamic model to assess the cost-effectiveness of the routine 4vHPV vaccination and the routine 4vHPV vaccination plus catch-up vaccination, compared with no vaccination (screening only) in Thai population, which is very meaningful. There are two issues that need to be explained.

1. In HPV Vaccination Strategies and Characteristics, the author described “Based on the local experience with other vaccines in Thailand, we assumed that for the routine 4vHPV vaccination, 95% of all girls 11-12 years of age received the routine regimen. For the catch-up vaccination, it was assumed that 70% of females between 13-24 years were vaccinated in the first 5 years”, is there any corresponding data to support the assumed percentage of vaccination?

Authors’ Response: 

Thailand’s success in providing care and coverage for all produces strong immunization outputs with nearly 100% coverage for all vaccines in the schedule[10]. In 1997, the Thai government introduced the National Immunization Program (NIP) with just BCG and DTP. This commitment to immunization and a subsequent push for the provision of vaccine services saw a big increase in coverage. 

By 1985, MCV and OPV were added and coverage rates jumped. By 1995, these four vaccines had at least 90% coverage across the country. In 2013, they all reached 99% (see Figure in the respond to Reviewers letter). The immunization schedule is slightly longer with rubella and HepB included, but all vaccines, with the exception of the second measles dose, has 99% coverage. The second dose was added to the schedule later and is for schoolchildren, administered after other vaccines [10]. 

We therefore assumed that 95% of all girls 11-12 years of age received the routine regimen which was slightly lower that the success rate for the routine vaccination of Thailand. We additionally cited the reference in the manuscript. 

2. In Model Calibration, the author described “the cervical cancer recognition parameters were adjusted to obtain the observed cervical cancer incidence rate to cervical cancer mortality rate ratio (i.e. 21.55/11.22 ≈ 2.30) “, but “the corresponding crude cervical cancer mortality rate was reported as 12.7 per 100,000 of which approximately 73.8%, i.e. 9.37 per 100,000, was assumed to be attributable to HPV 16/18. “, is there a calculation error in this value?

Authors’ Response: 

A typographical error is corrected (From “21.55/11.22≈ 2.30” to “21.55/9.37 ≈ 2.30”. Thank you very much.

References

1. Ngorsuraches S, Nawanukool K, Petcharamanee K, Poopantrakool U. Parents' preferences and willingness-to-pay for human papilloma virus vaccines in Thailand. J Pharm Policy Pract. 2015;8(1):20-. doi: 10.1186/s40545-015-0040-8. PubMed PMID: 26199734.

2. Sajirawattanakul D, Krittin P. Govt urged to drop HPV vaccine plan. (Apr 09. 2012). The Nation Thailand. 2012.

3. Kruiroongroj S, Chaikledkaew U, Thavorncharoensap M. Knowledge, acceptance, and willingness to pay for human papilloma virus (HPV) vaccination among female parents in Thailand. Asian Pacific journal of cancer prevention : APJCP. 2014;15(13):5469-74. Epub 2014/07/22. doi: 10.7314/apjcp.2014.15.13.5469. PubMed PMID: 25041020.

4. Hussain R, Bukhari NI, ur Rehman A, Hassali MA, Babar Z-U-D. Vaccine Prices: A Systematic Review of Literature. Vaccines. 2020;8(4):629. PubMed PMID: doi:10.3390/vaccines8040629.

5. Jacobson S. Vaccine pricing: how can we get it right? Expert Rev Vaccines. 2012;11:1163-5. doi: 10.1586/erv.12.95.

6. Sharma M, Ortendahl J, van der Ham E, Sy S, Kim JJ. Cost-effectiveness of human papillomavirus vaccination and cervical cancer screening in Thailand. BJOG : an international journal of obstetrics and gynaecology. 2012;119(2):166-76. Epub 2011/04/13. doi: 10.1111/j.1471-0528.2011.02974.x. PubMed PMID: 21481160.

7. Bruni L, Albero G, Serrano B, Mena M, Gómez D, Muñoz J, et al. ICO/IARC Information Centre on HPV and Cancer (HPV Information Centre). Human Papillomavirus and Related Diseases in Thailand. Summary Report 17 June 2019. [Date Accessed 13 DEC 2020] (URL:https://www.hpvcentre.net/statistics/reports/THA.pdf). 2019.

8. Attema AE, Brouwer WBF, Claxton K. Discounting in Economic Evaluations. Pharmacoeconomics. 2018;36(7):745-58. doi: 10.1007/s40273-018-0672-z. PubMed PMID: 29779120.

9. Permsuwan U, Guntawongwan K, Buddhawongsa P. Handling time in economic evaluation studies. Journal of the Medical Association of Thailand = Chotmaihet thangphaet. 2014;97 Suppl 5:S50-8. Epub 2014/06/27. PubMed PMID: 24964699.

10. Coe M, Gergen J. “Thailand Country Brief ( August 2017)”. Sustainable Immunization Financing in Asia Pacific. (URL: https://thinkwell.global/wp-content/uploads/2018/09/Thailand-Country-Brief-081618.pdf) Access Date 14 DEC 2020. Washington, DC: ThinkWell.: ThinkWell.; 2017.

---

## [Decision Letter · Decision Letter 1]

11 Jan 2021

The Epidemiologic and Economic Impact of a Quadrivalent Human Papillomavirus Vaccine in Thailand.

PONE-D-20-29823R1

Dear Dr. Termrungruanglert,

We’re pleased to inform you that your manuscript has been judged scientifically suitable for publication and will be formally accepted for publication once it meets all outstanding technical requirements.

Kind regards,

Luca Giannella

Academic Editor

PLOS ONE

Additional Editor Comments (optional):

Reviewers' comments:

Reviewer's Responses to Questions

**Comments to the Author**

1. If the authors have adequately addressed your comments raised in a previous round of review and you feel that this manuscript is now acceptable for publication, you may indicate that here to bypass the “Comments to the Author” section, enter your conflict of interest statement in the “Confidential to Editor” section, and submit your "Accept" recommendation.

Reviewer #1: All comments have been addressed

Reviewer #2: All comments have been addressed

2. Is the manuscript technically sound, and do the data support the conclusions?

Reviewer #1: Yes

Reviewer #2: (No Response)

3. Has the statistical analysis been performed appropriately and rigorously? 

Reviewer #1: I Don't Know

Reviewer #2: (No Response)

4. Have the authors made all data underlying the findings in their manuscript fully available?

Reviewer #1: No

Reviewer #2: (No Response)

5. Is the manuscript presented in an intelligible fashion and written in standard English?

Reviewer #1: Yes

Reviewer #2: (No Response)

6. Review Comments to the Author

Reviewer #1: This is an interesting research, using a cost-effect model to measure the epidemiologic and economic impact of HPV vaccination free in Thailand. I believe your country will benefit from this.

After reading the author's response and the latest manuscript, I think this paper is acceptable. But there is a small problem that needs attention.

1. If the format of Table 2 can be changed to that of Table 3 and Table 4, it may be more coordinated.

Reviewer #2: (No Response)

7. PLOS authors have the option to publish the peer review history of their article (what does this mean?). If published, this will include your full peer review and any attached files.

Reviewer #1: No

Reviewer #2: No

---

## [Editor Report · Acceptance letter]

25 Jan 2021

PONE-D-20-29823R1 

The epidemiologic and economic impact of a quadrivalent human papillomavirus vaccine in Thailand. 

Dear Dr. Termrungruanglert:

I'm pleased to inform you that your manuscript has been deemed suitable for publication in PLOS ONE. Congratulations! Your manuscript is now with our production department. 

Kind regards, 

on behalf of

Dr. Luca Giannella 

Academic Editor

PLOS ONE